# Reflecting on Evil and the Devil in Pentecostal Theodicy

## Marius Nel 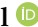

Unit for Reformed Theology, Faculty of Theology, Potchefstroom Campus, North-West University, Potchefstroom 2531, South Africa; marius.nel@nwu.ac.za

**Abstract:** Most Pentecostals, at least in the global South, believe that Satan and his demonic forces are responsible for much of the carnage and suffering that characterise life on earth. The broader context of the discourse is the challenge theodicy poses to Christian believers: if God is almighty and good, why do believers, just like unbelievers, suffer while living on earth? This paper aims to discuss Pentecostals' response: they blame evil as Satan's strategy to oppose God. They reason that his main goal is to handicap and double-cross creation because God threw him and his followers from heaven following his rebellion against the divine order. Thus, Satan is portrayed as the instigator of the first human couple, tempting them to sin, with all future generations implied by their choice and cursing human beings with a sinful nature at birth. Demonic forces employ human sinful nature to realise their ultimate goal, to separate humankind from the Creator by tempting them to sin while also threatening the rest of creation. This article investigates the Pentecostal theology of demonic forces in explaining the challenges posed by theodicy by comparing it to biblical data, using grammatical-historical exegesis and a comparative literature survey to evaluate their biblical grounding. It concludes that Pentecostals' belief in such forces are justified if the the New Testament narratives are accepted as divine revelation. However, some Pentecostal speculations about demons' origins, scope and reach are not biblically justified and complicates the response to theodicy, such as the origin of evil and its relation to human beings.

**Keywords:** theodicy; evil; Satan; demons; Genesis 3

## 1. Introduction

In reflecting on evil in theodicy from a Pentecostal hermeneutical perspective, several models qualify to present an explanation. Some scholars prefer first, ontological and/or theological models that explain evil as either intrinsically (ontologically) woven into the fabric of the universe or as being the result of God's (at least permissive) will for the world (Yong 2011, p. 162). The freewill model views evil either as the result of creaturely freedom unleashed by the fall of humankind (Gen 3) or the primordial fall of angels. God per se is not responsible for evil but created a world of free creatures who can choose to commit good or evil acts, with good or evil consequences (Yong 2011, p. 163). Second, the so–called "soul–making theodicy" argues that evil is allowed by God because of its formative capacities for the development of moral virtues, especially in terms of but not exclusive to believers. Evil is explained as beneficial for its soul–shaping outcomes (Yong 2011, p. 165). Amos Yong refers in the third place to a cruciform theology that states that God enters the world's suffering in the cross of Jesus Christ and the current suffering of the individual. Although it does not (and cannot) explain the origins of evil, it admits to the intractability of the problem of evil. At the same time, it insists that God is not removed from humans but has entered and embraced their suffering in God's own life.[1]

This article asks the question, what role do God's opponent and adversary, the antigod power of evil, and the evil one(s) play in the existence of evil in the world that affects, among others, human beings' well-being? Should Satan be blamed for the evil committed by and to people? Most Pentecostals, at least in the global South, link evil and the suffering that follows in its wake to Satan and demonic powers. Their link is based on their literalistic

reading of biblical texts, including narratives that are utilised to define their doctrines. At the same time, the devil's origin is taken back to the narrative of the Garden of Eden in Genesis 3. Before discussing this narrative, it is necessary to describe Pentecostals' view of Satan and his role in theodicy in more detail.

## 2. Satan and Theodicy

Although Pentecostals (and other Christians) blame Satan for the woes in the world, the brokenness that characterises nature and the suffering of all life forms, several of their cherished beliefs are not related to the viewpoints and different perspectives found in the Bible.

The first such belief is that Satan exclusively refers in the Bible to a supernatural person opposing God (and hence, the good). "Satan" is a Hebrew word that translates as "the one who opposes, obstructs, or accuses" (Sweet 1915, p. 2693). The Greek term literally means "adversary." Sometimes, the New Testament uses the term as a title or a name: (the) Satan. The Old Testament uses "satan" as a noun to refer to a human opponent, and it is usually translated as "adversary" or "enemy" (Seal 2016). For instance, when he built the temple, King Solomon stated that it was a good time to do so because he did not have any "satan" or "adversary" that might want to hinder him from this endeavour (1 Kg 5:4). David refers to Abishai in the same terms as an opponent or adversary (2 Sam 19:22). However, at times, the word "satan" also refers to a supernatural figure, as in Numbers 22:22, where an angel of YWHW is called a "satan" because he obstructs Balaam and his donkey's way ("stood in the road as an adversary to him"). That does not imply that the angel is a "satan," only that he is opposing Balaam in his purpose to thwart God's will. The Book of Job also refers to a supernatural "satan" figure without designating the figure's name. This entity functions in God's court, reports to God and can clearly only act in conjunction with God's wishes when he tests Job's faithfulness (Job 1:6). The term is used with a definite article, suggesting that the term does not refer to a specific person but rather to a title or an office. The entity functions on the same level as an angel ("messenger"). As a participant in the divine council along with the sons of God, it seems that his task is concerned with observing human beings on earth and serving as a messenger doing God's wishes among human beings. "The satan" functions as an accuser, not as a rebellious being undermining God's good purposes, as many Pentecostals suppose. Zechariah, in a similar fashion, refers to the "satan" (with a definite article, clearly to be translated as "the adversary" or "the accuser") within the context of a trial (Zech 3:1). The high priest Joshua stands before the Angel of the Lord. He is being accused by "the satan." "The accuser" is not acting maliciously; he serves as a prosecuting attorney, seemingly questioning without accusation whether Joshua is fit for the priesthood. In 1 Chronicles 21:1, "satan" (without definite article) is also connected to some supernatural being (cf. 2 Sam 24:1). The text creates the impression that it is the Lord that incites David by using a "satan." Clearly, "the satan" does not refer to the devil that Christians refer to, portrayed in later literature as God's chief rival, but to an office or function that falls under God's direct control. In the Old Testament, nowhere does one find a presentation of the "satan" as the personification of evil per se.

There might be two passages (Isa 14:12–15; Ezek 28:11–19), used repeatedly in popular Pentecostal literature, that seem to describe Satan's rebellion, as it is probably further interpreted by the *Life of Adam and Eve*, that originated in the first century BCE or CE. The text analyses the Daystar's fall (or Lucifer) as the fall of Satan and his angels (*Life of Adam and Eve* 12–15; see *2 Enoch* 29:4–5). Several passages also refer to angels as stars (Jud 5:20; Job 38:7; *1 Enoch* 104:1; *Testament of Moses* 10:9). According to the *Life*, Satan is a part of those creatures God cast out of heaven because of their rebellion in this tradition.

Second Temple literature (second and first centuries BCE) also applies the references to "satan" as a proper name (in *The Assumption of Moses* 10:1; *Jubilees* 2:23–29). This figure is the archenemy of God and humanity, linked in the *Wisdom of Solomon* with the phenomena of death (*Wisdom of Solomon* 2:24). His aim is death; for that reason, Satan had to cause human beings to sin that separates them from the source of their life, God (Wright 2006,

p. 19). Satan is also called Belial, Mastema and Satanail (*Testament of Gad* 4:7; *Testament of Benjamin* 7:1; *Testament of Reuben* 4:7; *Testament of Simeon* 5:3; *Testament of Asher* 3:2; *Jubilees* 11:5; *2 Enoch* 31:6). Believers verbalise the hope for Satan's speedy destruction as the author of evil and sin (*Testament of Levi* 18:12; *Testament of Judah* 25:3).

The New Testament connects to the figure that developed in the intertestamental period, viewing Satan as the primary enemy of God and believers. It is Satan who directly opposes Jesus from the start of his life, with Matthew 1:13 narrating how an angel warns Joseph to flee to Egypt to save the baby's life. Before beginning with his ministry of proclaiming the coming of the kingdom of God, Jesus is tested in the wilderness by Satan, a place of desertion and death connected to Satan (Mark 1:12–13; Matt 3:13–17; Luk 3:21–22; Joh 1:29–34). By winning over Satan by relying on what the Bible teaches (Matt 3:17; 4:1–11), Jesus obeys God's will, symbolically undoing Eve's and Israel's disobedience.

The New Testament describes several cases of individuals who are afflicted with demons. For instance, Luke 13:11, 16 connects an ill woman with "a spirit that had disabled her" with the words, "whom Satan bound for eighteen long years." Whether "Satan" in 13:16 and "spirit" in 13:11 refer to the same entity is unclear. John relates Judas' betrayal of Jesus to "Satan" who entered him (John 13:27). Does this refer to possession by Satan or oppression?[2] The text does not make it clear. According to Jesus, the devil is a busy enemy, trying to undermine the arrival of the kingdom of God. He is the evil one that snatches away the seed that falls along the path (Matt 13:19; Mark 4:15), causing people to neglect the message of the kingdom of God, and wicked people are called followers or children of the devil (John 8:44; Acts 13:10; Rev 2:9; 3:9; 1 John 3:8).[3]

However, Satan is also viewed as a defeated enemy (Matt 13:36–43; Rom 16:20; Heb 2:14–15; 1 John 3:8; Rev 20:2); the enemy is already conquered (Rev 12:7–10). However, that does not deny that Satan is aggressively attacking people and threatening believers' faith; several texts encourage believers to be on guard against his attacks (2 Thess 3:3; Jas 4:7; Rev 2:24).

Satan is called a dragon (Rev 20:2), a serpent (Rev 12:9), the evil one (John 17:15; Eph 6:16) and a tempter (Matt 4:3; 1 Thess 3:5) that prowls like a lion (1 Peter 5:8), characterising him as the enemy of God.[4] He is a ruler of the kingdom of the air and leader of the demonic realm (Eph 2:2). The belief aligns with and accommodates the first century CE's cultural beliefs that view spirits existing somewhere between heaven and earth. Satan is also referred to as "Beelzebul" (e.g., Matt 12:27; Luke 11:18), "lord of the house", or "lord of the heights". Perhaps it implies that Beelzebul or Satan is in charge of the demons.

Whence does Satan come from? According to Revelations 12:7, Satan lost during a war in heaven at the genesis of the world, and Michael and his angels cast out the devil and his angels from heaven (Rev 12:7). First John 3:8 describes the devil as one who "has been sinning from the beginning." The phrase "from the beginning" might refer to the Genesis account, linking the devil with the serpent, although it is improbable and implausible. The New Testament suggests that the "great dragon", or "the ancient serpent", who is called "the Devil and Satan, the deceiver of the whole world", was cast from heaven, that is, God's presence, down to the earth along with his followers, who were also angels. His fall was due to pride (1 Tim 3:6); he is a "murderer from the beginning", devoid of any truth (John 8:44) (Seal 2016).

Two dangers in terms of the devil's existence are, first, the assertion that such an entity does not exist except in the imagination and mythology of ancient people, seeing evil as a problem that requires a rational solution and, in second place, taking the devil too seriously, making him an easy scapegoat for people who are unwilling to take responsibility for their behaviour (Castelo 2012, p. 54). Especially among Pentecostals, the last danger prevails. The concept of spiritual warfare led many Pentecostals to actively pursue tactics that they assume assist the case of God. In the process, many of their efforts serve the unintended purpose of overemphasising Satan and his power. What should not be forgotten is that the war was already won on the hill of Golgotha, and believers are part of the cleanup process.

It cannot be denied that the New Testament argues in terms of a provisional cosmic dualism that consists of good and evil powers, in line with apocalyptic expectations, and that threaten God's purposes with creation. Christians recognise it and experience the tension of being involved in a battle between the forces of light and darkness, with their own choices, decisions and behaviour contributing to the good of tolerance and recognising the dignity of all people or the evil of decay and chaos.

## 3. Genesis 3's Influence on Theodicean Thinking

### 3.1. The Serpent in Genesis 2–3 and Satan

Most Pentecostals accept that the serpent in the Garden of Eden is Satan or even a creature under the direction of Satan, without any clear evidence in the Hebrew Bible. In the New Testament, Revelation 12:9 describes Satan or the Devil as the "ancient serpent" (without referring to the Eden episode) and the deceiver of the whole world. The book of *Wisdom of Solomon* (ca. 220 BCE–50 CE) might be the earliest literature to make this connection (2:24), also attested in *The Life of Adam and Eve* (33) and *2 Enoch* (31), all of them stating that was led Eve astray by the devil (Orlov 2012, p. 149).

It is submitted that the narrative in Genesis 3 instead presents a symbolic explanation of the origins of evil in the world and why it persists. Its concern is interpreting people's propensity to betray their God–given capacity to choose for or against God by disobeying God's clear command. The narrative is not about two people's choice to eat a fruit that determines the fate of everyone else, as the Augustinian-Calvinist doctrine of original sin asserts. Instead, the serpent symbolises the human inclination to choose evil, or rather, some elements called "sin" in the Bible. It explains that this choice results in some humans deliberately setting themselves up as rivals of God, illustrated in Adam and Eve hiding from the divine encounter, realising the consequences of their decision to taste the fruit (Gen 3:8). Genesis 1–2 explains the Creator's intention to design human beings to relate to the divine. However, to equate the serpent and Satan, as many Pentecostals do, cannot be justified from the evidence found in Genesis 3.[5]

Some other scholars subscribing to the concept of original sin do not accept that it is literally true. The idea instead describes the reality of the human condition, that the world is not an ideal place for all people, and people do not always act in a good, reasonable, or humane manner. For instance, Reinhold Niebuhr calls "original sin" both nations' and individuals' inherent urge to identify their own behaviour and view of what is right or good with what is universally right and good (Patterson 2023). In contrast, Paul Tillich argues that original sin refers to the sense of "alienation" human beings experience when living apart from their essential nature. As a result, they experience that they are trapped in what feels like a vast spiderweb, and their thrashing around makes matters only worse (Riggs 2023). The implication is that they need something or someone from outside themselves to save them, which the Bible calls "grace" and expresses in diverse ways, signalling God's benevolence in response to "original sin," should it be possible to accept the doctrine (Cox 2015, p. 29). Another train of thought comes from Jewish rabbi Harold Kushner, who argues for a God who hurts with hurting people, implying that God cannot be omnipotent. There are certain things that God cannot do, such as prevent people from committing evil acts that hurt themselves and those around them (Kushner 1993).

The Eastern Orthodox tradition rejected the doctrines of original sin, total depravity and hereditary guilt and argued that sin originated with the devil, the one who sins from the beginning (1 John 3:8). It argues that sin is not concerned with transmitted guilt but mortality, shared with the first couple (Gen 3:19c). Adam's punishment caused humans' inclination to sin without sharing in his responsibility. Instead of "original sin," Eastern Christians refer to "ancestral sin," implying that sin is hereditary, starting with Adam and Eve's tendency to sin that is passed on to their descendants, like water flowing from an infected source that also infects descendants. Although human nature is fallen or depraved, it is not totally the case, as Johan Cassian (c. 360–435 CE) asserted (Cassian 2023, p. 655). People have moral freedom and can choose to follow or disobey God. Although sparks of



goodwill still exist, any attempt to impress God is inadequate. Only divine intervention ensures salvation (Hassett 1913).

It is submitted that each person is responsible for their own choice to sin, although they share a tendency with all humanity to sin.[6] Evil did not originate with two people's foolish decision to disobey God. The Bible does not state where the human preference for evil comes from but emphasises that people can choose to do good or evil.

However, many Pentecostals interpret Genesis 3 literally, despite the troubling nature of the argument that Eden signifies the predetermined fall of all of humanity into sin, as the original doctrine of original sin asserts (Evans 2021, p. 2). Augustine formulated the doctrine of "original sin" due to Adam and Eve's disobedience, which led to evil and brokenness. Annette Evans explains that the doctrine implies that all human beings are born sinners without ever acquiring the ability to overcome their sinful nature. Such a teaching does not discount contemporary society's optimism about humanity's prospects to improve their character. Another implication is that believers struggle with the meaning and justice of inherited guilt and its proposed consequences and punishment. It is also not easy to reconcile concepts of human freedom and responsibility with a doctrine that states that humans are trapped in sin by their very nature and cannot help themselves.

What does the doctrine of original sin consist of, and can it be salvaged to make sense to contemporary people?

### 3.2. Augustine's Doctrines of Original Sin and Humanity's Total Depravity

Augustine (13 Nov 354-28 Aug 430 CE), the bishop of Hippo Regius in Numidia in Roman North Africa, reflected on the origins of evil. He argued that the creator God sent the Son to save human beings from evil and its results. But how can the world shaped by such a Creator include so much natural and moral evil? To justify God, Augustine necessitated three kinds of moves (Srigley 2019, pp. 53–68). In the first place, he argued that Eve's sin led to the advent of evil, the origin of all evil humans suffer. Their suffering is their just punishment because of original Adamic sin. Secondly, moral evil compromised human free will; Adamic sin implies that humans can never again be able to choose the good in themselves. Lastly, it is impossible to explain evil because it forms an essential element of the unplumbable mystery of God's unfathomable will, explicated by Paul's song about the divine unfathomable separation of the chosen elect from the rejected (see., e.g., Rom 9:18 "he has mercy on whomever he wishes, and he hardens whomever he wishes").

Augustine was responsible for formally systematising and developing formally the doctrine of original sin (*peccatum originale*) that had emerged in the third century CE. It attempted to explain why people are born depraved without the hope of consistently doing good due to their sinful nature; all people are born with an inherited tainted nature and a proclivity to sin. The Council of Trent formalised the doctrine in the sixteenth century; it is still accepted in the Roman Catholic tradition. Many Protestants also accept the doctrine's validity or a part of it, following Reformers like Martin Luther and John Calvin. Luther and Calvin maintained that original sin is connected to "concupiscence" (hurtful, evil desire). In other words, all people lost their free will except for sin. Nothing can erase original sin except the individual's choice to turn to Christ (Catholics baptise babies, trusting that it imparts grace and erases original sin to prevent the dying baby from being doomed; Protestants baptise infants because they are born into a family that is a part of the divine election).

Lutherans and Calvinists add that only the elect can accept Christ as Savior; the Spirit will commence to turn them to Christ. As a result, their sinful nature is weakened, although they will still be inclined to do evil, requiring them to fight against sin. Without Christ, people are predisposed to sin, taking shape in, *inter alia*, genocide, war, cruelty and other animalistic, cruel actions.

"Original sin" refers to humanity's inborn propensity to sin. Reformed theologian R.C. Sproul refers to Psalm 51:5, "Indeed, I was born guilty, a sinner when my mother conceived me" (Sproul 2017). In other words, people are not sinners because they sin;

they sin because they are (born) sinners. Not all people are utterly depraved, but all are totally depraved. Adam and Eve's fall captured human nature, eventually leading to illness and death. Sproul writes that people have become radically corrupt, affecting the roots of human lives (the Latin term for "root" is *radix*). People are essentially sinful.

Humanity's inborn propensity to sin due to Adam and Eve's disobedience led to the curse stated in Genesis 3:14–19. However, that is not clear from the text. It only says that Eve will experience pain during childbirth and will be subjected to her husband while the ground is cursed for Adam.[7] It does not refer to the reason for their offspring's fondness for sin and does not explain why a perfect divine creation contains so much evil. Original sin refers to a condition of being naturally inclined to sin, not something that people do or do not do. The Old Testament does not ever refer to the events in Genesis 3 again.

If human beings are, at their core, sinful, more than some minor adjustments or behavioural modifications are required. Only the Spirit's radical renovation from the inside can regenerate and quicken the believer. What is needed is that the Holy Spirit changes the human core, the heart.

Before attempting to design a theology of evil, it is essential to explain what the source of evil is and what its nature consists of.

## 4. Source of Evil

The question about the source of evil has intrigued humans from the beginning (Cox 2015, p. 24). The Priest depicts a God–shaping–chaos scenario (Gen 1:1-2:4a), while the Jahwist views creation from nothing (2:4b–5). The first account does not explain where evil came from, and the second does not answer whence the present disorder and evil came from and who created the "formless void" (Cox 2015, p. 24).

A later tradition in the Hebrew Bible proposes a personified evil force opposing the Creator and deceiving human beings (Bell 2007). The New Testament calls this adversary Satan or the devil, a roaring lion" who "prowls around, looking for someone to devour" (1 Pet 5:8). Many believers use other texts that do not refer to Lucifer or Satan (e.g., in Ezek and Job) to depict this figure's genesis, as God's second–in–command and chief among the angels who organised a heavenly rebellion. God used him to test and tempt God's children (the satan of Job 1:6–12). After his expulsion from heaven, he intended to undermine God's good creation.

Traditionally, theologians explained the challenge of evil as an essential part of theodicy that provides God's (actual or possible) reasons for permitting evil. Other arguments conclude that, as far as we know, no reason would justify God in permitting horrors, which represents sceptical theism. Other theologians admit that the problem of evil is a special difficulty for theism and impossible to explain. Still, they find adequate reasons or warrants for theism in natural theology and/or religious experience. Some argue that, despite theism not explaining the presence of evil adequately, its explanation is better than their metaphysical rivals' answer. Other scholars argue that the myth of Satan's shelf–life has expired without attempting.to explain the tragedy of a severely beaten and abused child, a mass murderer's coy smile, the Holocaust or the socialist Gulag's death camps. While human evil has always been central to religious thought, such discussions are conspicuously absent in the part of the church that caters for postmodern human beings, such as third–wave Neo–Pentecostals.

However, in the Third World, most believers experience the interference of Satan and his evil powers in continuation with their traditional cultural-religious traditions. Their experience is that when a "spirit" manifests, it typically identifies itself as an agent of the devil or Satan. As a result, many churches offer deliverance from evil spirits and curses. But if the myth is accepted that Satan rebelled against God, why did God not destroy him at the moment of his uprising? The question should be answered in conjunction with the question concerning evil among human beings.

Why does God allow evil to exist among people without eradicating it when it first manifested? The standard theological argument is that God created human beings with

free will, limiting divine power to interfere in human affairs to a certain extent. Why evil forces continue to exist, however, cannot be answered from any biblical data. The fact is that the Creator did and does not destroy evil.

## 5. Nature of Evil

What is the nature of evil in all its seriousness and ugliness? Historical events illustrated that the radical nature of evil and sin and their effects on human lives are to be taken seriously (Kärkkäinen 2002, p. 180).

In discussing human evil, theology must dialogue with psychology and psychiatry. However, psychology and psychiatry are challenged because the acknowledgement that evil exists outside humanity requires recognition that "supernatural" powers and forces exist. It is not possible to design experiments to ground such forces to normal forms of experimentation necessary to qualify for scientific endeavours. Because scientific inquiry requires clear evidence before examining a phenomenon, evil cannot be observed except by comparing its results and implications. Theology finds the source of knowledge about the existence of evil in "revelation", implying that it cannot make testable predictions, suggest and apply experiments, cannot be falsified and is, therefore, scientifically of no value. Believers rely on an intuitive sense of certainty that divine revelation is correct and does not follow the long and laborious route through reason, evidence, and hypothesis, the scientific path to truth. Some scientists call it a short-circuit certainty (Wathey 2022, p. 27).

On the other hand, the psychology of evil also cannot exist without a theology of evil, as Peck argues (Peck 1983, p. 50). If it does not incorporate valid insights from different religious traditions, its insights into the source and essence of evil are flawed.[8] Christians' alternative is required to order and provide meaning for human life in the face of radical evil.

Traditional Christian theology, functioning within the Platonic tradition, distinguished between moral evil and suffering. It defined moral evil (*malum culpae*) as the inevitable result of human (and angelic) free will. At the same time, it ascribed suffering not due to humans (*malum poenae*) but to malevolent spirits permitted by God to punish, correct or warn people (Freddoso 2001, p. 1). It did not deny God's omnipotence but explained that it did not include the operation of malevolent powers. Erich Fromm then defines moral evil as the desire to control other people to further one's interests that fosters imbalanced dependence on someone, discouraging their capacity to think for themselves and diminishing their originality. The purpose is to keep them in line and manipulate them to serve the evildoer's interest. The evil person tries to avoid the consequences of their behaviour by manipulating others to obey them like automatons and, in the process, robs them of their humanity (Fromm 1964, p. 124). Conversely, a good person appreciates other people and considers their unique needs and distinct personalities. Goodness results in life for all participants, while evil kills (Peck 1983, p. 42).

## 6. Conclusions

It is necessary to define evil and suffering further by reflecting on God's goodness, love and power. While philosophy defines evil classically as the absence of the good (*privatio boni*), corruption and perversion of the good, like darkness as the absence of light and silence of the lack of sound (Scott 2015, p. 13), sin can be defined as "a transgression of the righteous law of God" defines".[9]

Nevertheless, Pentecostals' definition of sin does not clearly distinguish between sin and evil. It might be better to define it as what was meant not to be, as the Creator did not intend it to be. Evil can then be divided into "natural" (suffering arising from natural causes), "moral" (suffering arising from human causes), and "metaphysical" evil (suffering arising from basic facts about the given human situation). Our discussion is limited to moral evil and suffering in its wake, which is theologically the most troubling aspect of evil.

In contrast to the New Testament, the Hebrew Bible introduces God as the creator who decided the fate of creation. All other forces are subject to God. Human beings can

choose between good and evil by endowing humankind with free will because they are created in the divine image. Evil falls within the range of God's reign and does not exist independently of God (Buber 1953, p. 94). Inherently, a human's malicious and perverted will does not exist as a substance but as a perversion of what is inherently good. Evil is then the abuse of the freedom God granted humans to choose (cf. Matt 19:8; Rom 5:12).

In contrast, the New Testament's diabolic dualism regards evil as being not God's creation but a dreadful cancer that God does not or cannot control. For instance, Paul writes that humans are under Satan's control because "they have participated in the sin of Adam" (Bell 2007, p. 256), and those "participating in the death and resurrection of Christ" have been "released from Satan's bondage" (Bell 2007, p. 263). Baptism and eucharist illustrate it ritually as "speech–event[s]" affecting "existential displacement" (Bell 2007, p. 279). However, Christ's victory must be affirmed daily by battling with evil. Christians' enemy is the devil, the evil rulers and authorities of the unseen world, mighty powers in this dark world, and evil spirits in the heavenly places (Eph 6:11–12), and to counter their strategies, they should put on the whole armour of God (Eph 6:13).

The question can be asked: if the creator created human beings with the capacity to choose between good and evil, did God create an imperfect world? Is God at least partly responsible for the human choosing for evil by leaving them the freedom to choose? Was God's creation of human beings a flaw?

It is related to asking if and to what extent God is involved in the daily lives of humanity. Does God cause everything that happens? If answered positively, it implies that what happens is related to God's choice. In that case, a natural disaster or sickness indicates that God wants to realise a certain goal. However, determinism in justifying God's actions is not supported by biblical evidence or daily experience. Brokenness, the state of entropy of all living things, the existence of bacteria and viruses and the occurrence of hatred, strife and conflict characterise creation.

It might instead be faithful to biblical evidence that God, the primordial reality, is the primary cause of what happens because the creator is responsible for its existence. It includes evil. At the same time, human beings, as moral agents, determine their own lives to a certain extent through their choices. Sometimes, people's talents and capacities are genetically determined, but it remains true that they decide how they apply their talents and capabilities.

Christians find the source of goodness in the creator, a good God. Moral evil cannot be associated with God and was not created. Instead, it is the perversion of creation, wrong choices, and actions that are evil as the result of people's abuse of their divinely ordained freedom. By acting evilly, people choose anti–good and anti–God. While the Pentecostal tradition blames demonic forces for evil manifestations among people, it is submitted that human culpability in evil acts should be emphasised.

Sin can become a collective endeavour, seen in the exercise of racism, sexism and genderism by larger groups of people and based on the stereotyping of people as "the other." In conservative religious circles, as in Pentecostalism, it occurs, *inter alia*, in the rejection of people with different sexual orientations who are typified as "sinners." It results in structural or systemic sin based on shared prejudices (Wright 2006, p. 19). In considering theodicy, it should be remembered that much of the evil humans suffer is related to human choices contrary to their divine purpose.

**Funding:** This research received no external funding.

**Institutional Review Board Statement:** Not applicable.

**Informed Consent Statement:** Not applicable.

**Data Availability Statement:** No new data were created or analyzed in this study. Data sharing is not applicable to this article.

**Conflicts of Interest:** The author declares no conflict of interest.

## Notes

[1] ([Yong 2011](#), pp. 165–66). The same distinction (and others) is found among other scholars; see Yong's references.

[2] Most translations translate δαιμονιζομένος as "demon-possessed" rather than "demon-obsessed." Later, some scholars distinguished between obsession as a state that can affect Christians while possession refers to people of evil without any faith relation. However, the New Testament does not make such a distinction. Cases of δαιμονιζόμενον mentioned in the Gospels include Mark 1:32; 3:22; 5:15, 16, 18; 7:25; Matt 4:24; 8:16, 28, 33; 12:22; Luke 8:36 and John 10:21.

[3] Mark 4:14 refers to Satan, but the synoptic parallels each use a different word for Satan here: Matt 13:19 has "the evil one," while Luke 8:12 has "the devil," illustrating the fluidity of the gospel tradition in often using synonyms at the same point of the parallel tradition.

[4] The connection between "Satan" and the serpent is discussed in the next section.

[5] The only reference in the New Testament to such an equation, although highly improbably, is in Rev 20:2: "He seized the dragon—the ancient serpent, who is the devil and Satan—and tied him up for a thousand years".

[6] Even mentally challenged people share the inclination to do evil, although they cannot be held responsible or accountable for their actions.

[7] https://www.bbc.co.uk/religion/religions/christianity/beliefs/originalsin_1.shtml; accessed on 8 July 2021.

[8] That psychology and psychiatry sometimes do not know what to do with evil is demonstrated by the four volumes of *Psychology and the Bible*, edited by Ellens and Rollins (2004), that discuss psychology as an angle to read the Bible. No reference to evil can be found in these volumes.

[9] The Westminster Confession Chapter 6.6. https://www.ligonier.org/learn/articles/westminster-confession-faith/; accessed 21 June 2021.

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
