# Peer review of "Reflecting on Evil and the Devil in Pentecostal Theodicy"

_religions, doi:10.3390/rel15040483_

Round 1

Reviewer 1 Report

Comments and Suggestions for Authors

No comments with regard to the content. The article is very well-written and responds to a very relevant issue.

There is only a formal aspect that the author should consider. I'm afraid the references should be rearranged. If I'm not mistaken, according to the Style Sheet, Endnotes should be removed and in-text citations should be included instead.

Author Response

Reviewer 1

No comments

Reviewer 2

Comments and Suggestions for Authors

Title

Although this is a well crafted title, I am not too sure with the significance of 'Some Critical Remarks'? is since offering critical remarks is precisely what is expected with every paper. Furthermore, to the extent that not many pentecostal scholars have adequately defined 'Pentecostal Theodicy', its inclusion in your title will lead to anticipation of its exploration or contextualisation within constructive/systematic theology

Suggestion followed. 

Abstract 

In your abstract 'the discourse is the challenge theodicy poses to Christian believers', must be accopanied by the specific theodicy being referred to .. Person making theodicy, process theodicy, or which exactly? 

The followup sentence: 'Demonic forces employ human sinful nature to realise their ultimate goal, to separate humankind from the Creator by tempting them to sin while also threat- ening the rest of creation' is really not necessary in your abstract ..... Its meaning is already captured in the preceding sentence. 

Suggestion followed. 

Please shorten the following projected conclusion in your abstract: 'It concludes that Pentecostals’ belief in such forces may be justified if the narratives in the New Testament are accepted as divine revelation but that some of the Pentecostal speculations about the origins, scope and reach of demons are not biblically justi- fied' - try to make it less wordy so that your readers may follow its logic in relation to your broader argument

Suggestion followed.

The following is not necessary, especially to the extent that conclusion had been projected. So, please review or decide on the relevance of: It complicates the response to some of the challenges of theodicy, such as the origin of evil and its relation to human beings'..... 

Please make sure that your Abstract has the following core elements which are currently lacking: 

  1. Clearly describe the main aim of this paper, particularly in relation to or in the mirror of recent theodicies 
  2. Indicate the academic and practical importance of this paper in the field of theology and its contribution to the broader pentecostal scholarship. 
  3. Please describe the the methodology used: it is already provided: using grammatical-historical exegesis and a comparative literature survey to evaluate their biblical grounding.
  4. Preface your projected conclusion with summary of finding(s).

Suggestion followed.

Introduction

Please insure that the greater body of your introduction aligns with the logic of your title, abstract, introduces sub themes and even aligns with your conclusion (which is currently not there) -- Introduction, Abstract and Conclusion constitute the coherent functionality of an academic paper, and as such none must be excluded. 

Suggestion followed

Furthermore, your introduction does not say anything about 'Pentecostal Theodicy' as projected in your title.. please correct this accordingly by ensuring that it becomes the heart of your introduction or the central theological framework upon which the argument of your paper is built. 

Suggestion followed

First Subtheme

Please introduce Pentecostal Theodicy as your primary submtheme and then show the manner in which your paper is going to either problematize or use it to make an academic argument.  Please make sure to cite available literature supporting it as a theoretical framework within the field of theology and particularly the pentecostal scholarship

Suggestion followed

Conclusion 

Please take note that every contribution must have Introduction, bodywork and specified conclusion. Thus, without a conclusion as it is the case with your paper, not much can be achieved in terms of aligning projected objectives with outcomes. Please be deliberate on the inclusion of conclusion as the last section of your paper. It cannot be expected that your readers will deduce what ought to be a 'conclusion' from reading your last subtheme dealing with theology of evil. 

Suggestion followed 

Comments on the Quality of English Language

The use of English Language and academic terms was excellent

Reviewer 3

Comments and Suggestions for Authors

It should be noted that the global south Pentecostals do not interpret the Bible allegorically but literally. One would think that a space should be given to their own theological resonations with evils because, if one uses the Enlightenment demythologised methodology, which many critics of global south Pentecostals have done, a crucial aspect of decolonisation might be missed. For instance, the Genesis story is taken as a real story rather than the Enlightenment’s categorization and interpretation of Genesis 1-11 as allegorical and unscientific. The global south Pentecostalism agrees with Augustinian-Calvinian conception of the original sin, and builds its theology on this premise.

Suggestion followed

The argument about the original sin does not give vent to the global south Pentecostalism, but it antagonises it with authorities and theologies from the global north. This is a problem of delegitimising the global south pentecostalism’s stand on their critical and contextual interpretation and application of the text. It appears that pentecostalism re-asserts patristic theology of original sin, rather than the Enlightenment interpretations. This should be examined.

Suggestion followed 

Finally, the last section of the paper does not situate itself within the Pentecostal thought, particularly with its eschatological teaching in which moral sin or failure will be judged by God. It would be helpful if the author lets the readers into how the pentecostals, rather than the Catholic and Protestant theologies and secular philosophies, deal with the types of evil discussed in the paper.

Suggestion followed

Reviewer 2 Report

Comments and Suggestions for Authors

Title

Although this is a well crafted title, I am not too sure with the significance of 'Some Critical Remarks'? is since offering critical remarks is precisely what is expected with every paper. Furthermore, to the extent that not many pentecostal scholars have adequately defined 'Pentecostal Theodicy', its inclusion in your title will lead to anticipation of its exploration or contextualisation within constructive/systematic theology

Abstract 

In your abstract 'the discourse is the challenge theodicy poses to Christian believers', must be accopanied by the specific theodicy being referred to .. Person making theodicy, process theodicy, or which exactly? 

The followup sentence: 'Demonic forces employ human sinful nature to realise their ultimate goal, to separate humankind from the Creator by tempting them to sin while also threat- ening the rest of creation' is really not necessary in your abstract ..... Its meaning is already captured in the preceding sentence. 

Please shorten the following projected conclusion in your abstract: 'It concludes that Pentecostals’ belief in such forces may be justified if the narratives in the New Testament are accepted as divine revelation but that some of the Pentecostal speculations about the origins, scope and reach of demons are not biblically justi- fied' - try to make it less wordy so that your readers may follow its logic in relation to your broader argument

The following is not necessary, especially to the extent that conclusion had been projected. So, please review or decide on the relevance of: It complicates the response to some of the challenges of theodicy, such as the origin of evil and its relation to human beings'..... 

Please make sure that your Abstract has the following core elements which are currently lacking: 

1. Clearly describe the main aim of this paper, particularly in relation to or in the mirror of recent theodicies 

2. Indicate the academic and practical importance of this paper in the field of theology and its contribution to the broader pentecostal scholarship. 

3. Please describe the the methodology used

4. Preface your projected conclusion with summary of finding(s).

Introduction

Please insure that the greater body of your introduction aligns with the logic of your title, abstract, introduces sub themes and even aligns with your conclusion (which is currently not there) -- Introduction, Abstract and Conclusion constitute the coherent functionality of an academic paper, and as such none must be excluded. 

Furthermore, your introduction does not say anything about 'Pentecostal Theodicy' as projected in your title.. please correct this accordingly by ensuring that it becomes the heart of your introduction or the central theological framework upon which the argument of your paper is built. 

First Subtheme

Please introduce Pentecostal Theodicy as your primary submtheme and then show the manner in which your paper is going to either problematize or use it to make an academic argument.  Please make sure to cite available literature supporting it as a theoretical framework within the field of theology and particularly the pentecostal scholarship

Conclusion 

Please take note that every contribution must have Introduction, bodywork and specified conclusion. Thus, without a conclusion as it is the case with your paper, not much can be achieved in terms of aligning projected objectives with outcomes. Please be deliberate on the inclusion of conclusion as the last section of your paper. It cannot be expected that your readers will deduce what ought to be a 'conclusion' from reading your last subtheme dealing with theology of evil. 

Comments on the Quality of English Language

The use of English Language and academic terms was excellent

Author Response

Reviewer 2

Comments and Suggestions for Authors

Title

Although this is a well crafted title, I am not too sure with the significance of 'Some Critical Remarks'? is since offering critical remarks is precisely what is expected with every paper. Furthermore, to the extent that not many pentecostal scholars have adequately defined 'Pentecostal Theodicy', its inclusion in your title will lead to anticipation of its exploration or contextualisation within constructive/systematic theology

Suggestion followed. 

Abstract 

In your abstract 'the discourse is the challenge theodicy poses to Christian believers', must be accopanied by the specific theodicy being referred to .. Person making theodicy, process theodicy, or which exactly? 

The followup sentence: 'Demonic forces employ human sinful nature to realise their ultimate goal, to separate humankind from the Creator by tempting them to sin while also threat- ening the rest of creation' is really not necessary in your abstract ..... Its meaning is already captured in the preceding sentence. 

Suggestion followed. 

Please shorten the following projected conclusion in your abstract: 'It concludes that Pentecostals’ belief in such forces may be justified if the narratives in the New Testament are accepted as divine revelation but that some of the Pentecostal speculations about the origins, scope and reach of demons are not biblically justi- fied' - try to make it less wordy so that your readers may follow its logic in relation to your broader argument

Suggestion followed.

The following is not necessary, especially to the extent that conclusion had been projected. So, please review or decide on the relevance of: It complicates the response to some of the challenges of theodicy, such as the origin of evil and its relation to human beings'..... 

Please make sure that your Abstract has the following core elements which are currently lacking: 

  1. Clearly describe the main aim of this paper, particularly in relation to or in the mirror of recent theodicies 
  2. Indicate the academic and practical importance of this paper in the field of theology and its contribution to the broader pentecostal scholarship. 
  3. Please describe the the methodology used: it is already provided: using grammatical-historical exegesis and a comparative literature survey to evaluate their biblical grounding.
  4. Preface your projected conclusion with summary of finding(s).

Suggestion followed.

Introduction

Please insure that the greater body of your introduction aligns with the logic of your title, abstract, introduces sub themes and even aligns with your conclusion (which is currently not there) -- Introduction, Abstract and Conclusion constitute the coherent functionality of an academic paper, and as such none must be excluded. 

Suggestion followed

Furthermore, your introduction does not say anything about 'Pentecostal Theodicy' as projected in your title.. please correct this accordingly by ensuring that it becomes the heart of your introduction or the central theological framework upon which the argument of your paper is built. 

Suggestion followed

First Subtheme

Please introduce Pentecostal Theodicy as your primary submtheme and then show the manner in which your paper is going to either problematize or use it to make an academic argument.  Please make sure to cite available literature supporting it as a theoretical framework within the field of theology and particularly the pentecostal scholarship

Suggestion followed

Conclusion 

Please take note that every contribution must have Introduction, bodywork and specified conclusion. Thus, without a conclusion as it is the case with your paper, not much can be achieved in terms of aligning projected objectives with outcomes. Please be deliberate on the inclusion of conclusion as the last section of your paper. It cannot be expected that your readers will deduce what ought to be a 'conclusion' from reading your last subtheme dealing with theology of evil. 

Suggestion followed 

Comments on the Quality of English Language

The use of English Language and academic terms was excellent

Reviewer 3 Report

Comments and Suggestions for Authors

It should be noted that the global south Pentecostals do not interpret the Bible allegorically but literally. One would think that a space should be given to their own theological resonations with evils because, if one uses the Enlightenment demythologised methodology, which many critics of global south Pentecostals have done, a crucial aspect of decolonisation might be missed. For instance, the Genesis story is taken as a real story rather than the Enlightenment’s categorization and interpretation of Genesis 1-11 as allegorical and unscientific. The global south Pentecostalism agrees with Augustinian-Calvinian conception of the original sin, and builds its theology on this premise.

The argument about the original sin does not give vent to the global south Pentecostalism, but it antagonises it with authorities and theologies from the global north. This is a problem of delegitimising the global south pentecostalism’s stand on their critical and contextual interpretation and application of the text. It appears that pentecostalism re-asserts patristic theology of original sin, rather than the Enlightenment interpretations. This should be examined.

Finally, the last section of the paper does not situate itself within the Pentecostal thought, particularly with its eschatological teaching in which moral sin or failure will be judged by God. It would be helpful if the author lets the readers into how the pentecostals, rather than the Catholic and Protestant theologies and secular philosophies, deal with the types of evil discussed in the paper.

Author Response

Reviewer 3

Comments and Suggestions for Authors

It should be noted that the global south Pentecostals do not interpret the Bible allegorically but literally. One would think that a space should be given to their own theological resonations with evils because, if one uses the Enlightenment demythologised methodology, which many critics of global south Pentecostals have done, a crucial aspect of decolonisation might be missed. For instance, the Genesis story is taken as a real story rather than the Enlightenment’s categorization and interpretation of Genesis 1-11 as allegorical and unscientific. The global south Pentecostalism agrees with Augustinian-Calvinian conception of the original sin, and builds its theology on this premise.

Suggestion followed

The argument about the original sin does not give vent to the global south Pentecostalism, but it antagonises it with authorities and theologies from the global north. This is a problem of delegitimising the global south pentecostalism’s stand on their critical and contextual interpretation and application of the text. It appears that pentecostalism re-asserts patristic theology of original sin, rather than the Enlightenment interpretations. This should be examined.

Suggestion followed 

Finally, the last section of the paper does not situate itself within the Pentecostal thought, particularly with its eschatological teaching in which moral sin or failure will be judged by God. It would be helpful if the author lets the readers into how the pentecostals, rather than the Catholic and Protestant theologies and secular philosophies, deal with the types of evil discussed in the paper.

Suggestion followed

Round 2

Reviewer 2 Report

Comments and Suggestions for Authors

I am very glad that the author followed most of my suggestions accurately since this has significantly improved the quality of this paper. Congratulations and all the best to the author for the remaining production process...